# Alpha-Glucosidase Inhibition and Molecular Docking of Isolated Compounds from Traditional Thai Medicinal Plant, *Neuropeltis racemosa* Wall.

**DOI:** 10.3390/molecules27030639

**Published:** 2022-01-19

**Authors:** Oraphan Sakulkeo, Chatchai Wattanapiromsakul, Thanet Pitakbut, Sukanya Dej-adisai

**Affiliations:** 1Department of Pharmacognosy and Pharmaceutical Botany, Faculty of Pharmaceutical Sciences, Prince of Songkla University, Hat Yai 90112, Songkhla, Thailand; oraphan.s@psu.ac.th (O.S.); chatchai.w@psu.ac.th (C.W.); 2Department of Biochemical and Chemical Engineering, Technical University of Dortmund, 44227 Dortmund, Germany; thanet.pitakbut@tu-dortmund.de

**Keywords:** *Neuropeltis racemosa*, alpha-glucosidase inhibition, anti-diabetes, phytochemistry, molecular docking

## Abstract

*Neuropeltis racemosa* Wall. (Convolvulaceae) is wildly distributed in Asia. Its stem is used as the component in traditional Thai recipes for treatments of muscle rigidity, skin disorder, dysentery, and hypoglycemia. However, the chemical constituents and biological activities of *N. racemosa* have not been reported. From a screening assay, *N. racemosa* stem crude extract showed the potent effect on alpha-glucosidase inhibition at 2 mg/mL as 96.09%. The bioassay-guiding isolation led to 5 compounds that were identified by spectroscopic techniques as scopoletin (**1**), syringic acid (**2**), methyl 3-methyl-2-butenoate (**3**), *N-trans-*feruloyltyramine (**4**), and *N-trans-* coumaroyltyramine (**5**). Compounds **1**, **4**, and **5** exhibited an IC_50_ of 110.97, 29.87, and 0.92 µg/mL, respectively, while the IC_50_ of positive standard, acarbose was 272.72 µg/mL. Kinetic study showed that compound **1** performed as the mixed-type inhibition mechanism, whereas compounds **4** and **5** displayed the uncompetitive inhibition mechanism. The docking study provided the molecular understanding of isolated aromatic compounds (**1**, **2**, **4** and **5**) to alpha-glucosidase. Hence, this study would be the first report of isolated compounds and their anti-alpha-glucosidase activity with the mechanism of action from *N. racemosa*. Thus, these active compounds will be further studied to be the lead compounds among natural antidiabetic drugs.

## 1. Introduction

Diabetes mellitus (DM) is metabolic syndrome related to a hyperglycemia condition. In 2021, The International Diabetes Federation reported number of diabetes adults is 537 million and the trend will rise to 783 million in 2045 [1]. Two major diabetic types are type 1 diabetes (insulin defection from the problem of beta cells) and type 2 diabetes (insulin resistance from the inability of body cells). Patients of both types have a blood glucose uncontrollable condition. Exogenous insulin is considered in patients with type 1 diabetes, while many classes of glucose-lowering agents are used for patients with type 2 diabetes [2]. Although many clinical compounds are available, drugs used in the management of DM are complicated [3]. Therefore, new drug development for diabetes is ongoing. Folk medicine is the alternative treatment that provides sources of interesting herbal medicine. The medicinal plants were investigated for their potential ability using in vitro bioactivity-targeted screening.

*Neuropeltis racemosa* Wall. (Convolvulaceae) is wildly distributed in Asian countries such as China, Myanmar, Malaysia, Indonesia [4], and including many regions of Thailand. The stem of *N. racemosa* is used as a component of traditional Thai recipes for treatments of muscle rigidity, skin disorder, dysentery, and hypoglycemia [5,6,7]. *N. racemosa* is also used as a component of Mathurameha (traditional Thai recipe) that has been reported in regards to hypoglycemic activity [6]. It agreed with our preliminary screening assay, as the methanolic extract of *N. racemosa* stem showed the potent effect of alpha-glucosidase inhibition at 2 mg/mL as 96.09%. Nowadays, the study of *N. racemosa* remains poor. Therefore, *N. racemosa* stem was selected for an investigation of chemical constituents, alpha-glucosidase inhibition, mode of action, and a molecular docking study.

## 2. Results

### 2.1. Determination of Alpha-Glucosidase Inhibition

The stem of *N. racemosa* was extracted with different solvent by increasing the polarity from *n-*hexane, EtOAc, EtOH, and water, respectively. Four solvent extracts were determined on alpha-glucosidase inhibitory activity. *n-*Hexane, EtOAc, EtOH, and water extracts exhibited alpha-glucosidase inhibition with an IC_50_ of 56.81 g/mL, 191.44 μg/mL, 39.65 μg/mL, and 4.02 mg/mL, respectively, while the standard, acarbose exhibited an IC_50_ of 245.95 μg/mL. Therefore, EtOH extract showed the highest inhibition effect towards the alpha-glucosidase enzyme.

### 2.2. Extraction, Isolation, and Identification of Pure Compounds

From the bioactivity-guided isolation of the *N. racemose* stem, the EtOH extract was selected to isolate pure compounds by chromatographic techniques. Five pure compounds (**1**–**5**) were isolated and identified as scopoletin (**1**), syringic acid (**2**), methyl 3-methyl-2-butenoate (**3**), *N*-*trans*-feruloyltyramine (**4**), and *N*-*trans*-coumaroyltyramine (**5**) (Figure 1).

#### 2.2.1. Scopoletin (**1**)

Compound **1** was obtained as yellow pale needles and dissolved in chloroform. The UV spectrum in chloroform showed an absorption maximum (λ_max_) at 341 nm. The IR spectrum demonstrated maximum absorption bands at 3460, 1651, 1218, 772, and 669 cm^−1^. The ^1^H-NMR spectrum of **1** exhibited six proton signals. The two doublet signals at δ_H_ 6.25 (1H, d, *J* = 9.5 Hz, H-3) and 7.58 (1H, d, *J* = 9.5 Hz, H-4) were determined as olefinic protons. The others are four singlet signals. Two of four singlet signals at δ_H_ 6.63 (1H, s, H-6) and 6.90 (1H, s, H-9) belong to aromatic proton. The signal at δ_H_ 6.17 (1H, s, OH-8) was determined as the hydroxyl proton, while the signals at δ_H_ 3.94 (3H, s, OCH_3_-7) was assigned as methoxyl proton. The ^13^C-NMR spectrum showed ten carbon signals that were observed: three oxygenated aromatic carbon signals at δ_C_ 143.96 (C-7),149.64 (C-8) and 150.23 (C-10); two tertiary aromatic carbon signals at δ_C_ 103.18 (C-9) and 107.44 (C-6); two olefinic methine signals at δ_C_ 113.42 (C-3) and 143.29 (C-4); one quaternary carbon signals at δ_C_ 111.48 (C-5); one carbonyl signal at δ_C_ 161.43 (C-2), and one methoxyl signal at δ_C_ 56.39 (OCH_3_-7). The molecular formula of **1** is C_10_H_8_O_4_. The HMBC correlations (Figure 2) were used for confirming the structure substitution. The NOESY experiment of **1** showed the correlation signal between the proton of C-6 and the methoxyl proton of C-7. As mentioned above, it was suggested that compound **1** is a scopoletin that is different from iso-scopoletin [8].

#### 2.2.2. Syringic Acid (**2**)

Compound **2** was obtained as an orange amorphous solid and dissolved in methanol. The UV spectrum in methanol demonstrated an absorption maximum (λ_max_) at 289 nm. The IR spectrum exhibited bands at 3434, 1634, and 1426 cm^−1^. The ^1^H-NMR spectrum of **2** exhibited three singlet proton signals. The singlet signal at δ_H_ 3.78 (6H, s, OCH_3_-3 and OCH_3_-5) was assigned as two methoxyl proton signals, while the singlet signal at δ_H_ 7.19 (2H, s, H-2 and H-6) was determined as aromatic protons. The last broad singlet signal at δ_H_ 9.07 (1H, br, COOH-1) was defined as the carboxylic proton. The HR-ESIMS showed an [M − H]^−^ ion peak at *m/z* 197.0456, and correlated with a molecular formula of C_9_H_10_O_5_. The structure of **2** was determined to be syringic acid.

#### 2.2.3. Methyl 3-Methyl-2-butenoate (**3**)

Compound **3** was obtained as a brown amorphous solid and dissolved in chloroform. The UV spectrum in chloroform demonstrated an absorption maximum (λ_max_) at 290 nm. The IR spectrum exhibited band at 2922, 1698-1595, 1457, 1260–1220, 1110, and 874–800 cm^−1^. The ^1^H-NMR spectrum showed three singlet proton signals at δ_H_ 1.2 (6H, s, CH_3_-3 and H-4), 3.8 (3H, s, OCH_3_-1), and 5.8 (1H, s, H-2). The ^13^C-NMR spectrum showed six carbon signals that were observed: one quaternary carbon signal at δ_C_ 157.30 (C-3); one olefinic methine signal at δ_C_ 107.42 (C-2); one carbonyl signal at δ_C_ 186.84 (C-1); one methoxyl signal at δ_C_ 56.48 (OCH_3_-1), and two methyl signals at δ_C_ 29.69 (C-4) and at δ_C_ 29.35 (CH_3_-3). The HMBC correlations (Figure 3) suggested the substitution of **3**. The structure of **3** was elucidated as methyl 3-methyl-2-butenonoate.

#### 2.2.4. *N*-*trans*-Feruloyltyramine (**4**)

Compound **4** was obtained as a white amorphous solid and dissolved in methanol. The UV spectrum in methanol showed an absorption maximum (λ_max_) at 318 nm. The IR spectrum demonstrated band at 3434, 1652, 1542, and 978 cm^−1^. The ^1^H-NMR spectrum exhibited the methoxyl proton signal at δ_H_ 3.78 (3H, s, OCH_3_-6). The two doublets at δ_H_ 6.39 (1H, d, *J* = 15.62 Hz, H-2) and 7.42 (1H, d, *J* = 15.62 Hz, H-3) indicated the presence of two trans protons. The two doublet of doublet signals at δ_H_ 3.45 (2H, dd, *J* = 7.08 and 7.56 Hz, H-1′) and 2.75 (2H, dd, *J* = 7.08 and 7.56 Hz, H-2′) were determined as methylene protons which conjugated to secondary amine and aromatic parts, respectively. The residue five proton signals at δ_H_ 6.72 (2H, dd, *J* = 9.27 and 2.40 Hz, H-5′ and H-7′), 6.78 (1H, d, *J* = 8.06 Hz, H-8), 7.01 (1H, dd, *J* = 8.29 and 1.95 Hz, H-9), 7.05 (2H, dd, *J* = 9.27 and 2.40 Hz, H-4′ and H-8′), and 7.11 (1H, d, *J* = 1.95 Hz, H-5) were indicated as seven aromatic protons of two aromatic rings. The ^13^C NMR showed sixteen carbon signals; however, they were indicated as eighteen carbons: carbonyl carbon at δ_C_ 169.19 (C-1); the methoxyl carbon δ_C_ 56.42 (OCH_3_-6); two methylene carbons at δ_C_ 35.81 (C-2′) and 42.53 (C-1′); two olefinic methine carbon at δ_C_ 118.79 (C-2) and 142.01 (C-3); two quaternary aromatic carbon at δ_C_ 128.31 (C-4) and 131.22 (C-3′); three oxygenated aromatic carbons at δ_C_ 149.32 (C-6), 149.88 (C-7) and 156.94 (C-6′); five tertiary aromatic carbons at δ_C_ 111.63 (C-5), 116.42 (C-8), 123.22 (C-9) 130.72 (C-4′ and C-8′), and 116.28 (C-5′ and C-7′). The HR-ESIMS showed an [M + Na]^+^ ion peak at *m/z* 336.1206 (calcd. for [C_18_H_19_NO_4_ + Na] ^+^), correlated with a molecular formula of C_18_H_19_NO_4_. The structure of **4** was determined to be *N*-*trans*-feruloyltyramine.

#### 2.2.5. *N*-*trans*-Coumaroyltyramine (**5**)

Compound **5** was obtained as a white amorphous solid and dissolved in methanol. The UV spectrum in methanol showed an absorption maximum (λ_max_) at 295 nm. The IR spectrum demonstrated a band at 3434, 1637, 1541, and 980 cm^−1^. The ^1^H spectrum of **5** showed doublet signals at δ_H_ 6.38 (1H, d, *J* = 15.86 Hz, C-2) and 7.31 (1H, d, *J* = 15.86 Hz, C-3) which were assigned as two *trans* protons. The doublet of doublet signals at δ_H_ 3.45 (2H, dd, *J* = 7.07 and 7.57 Hz, C-1′) and 2.75 (2H, dd, *J* = 7.08 and 7.57 Hz, C2′) were determined as two methylene protons. Other four proton signals, at δ_H_ 6.70 (2H, m, H-6), 6.78 (2H, m, H-5), 7.05 (2H, dt; *J* = 9.00 and 2.00 Hz, H-5′), and 7.38 (2H, dt; *J* = 9.00 and 2.00 Hz, H-8′), were assigned as protons on two symmetric aromatic rings. The ^13^C NMR showed thirteen carbon signals; however, they were indicated as seventeen carbons: one carbonyl carbon at δ_C_ 169.25 (C-1); two methylene signals at δ_C_ 35.82 (C-2′) and 42.53 (C-1′); two olefinic methine carbon signals at δ_C_ 118.49 (C-2) and 141.76 (C-3); two quaternary carbon signals at δ_C_ 127.77 (C-4) and 131.34 (C-3′); two oxygenated aromatic carbons at δ_C_ 156.92 (C-6′) and 160.50 (C-7); eight tertiary aromatic carbons at δ_C_ 130.3 (C-5 and C-9), 116.72 (C-5′ and C-7′), 116.28 (C-6 and C-8), and 130.72 (C-4′ and C-8′). The HR-ESIMS showed an [M + Na]^+^ ion at *m/z* 306.1101 (calcd. for [C_17_H_17_NO_3_ + Na] ^+^), and correlated with a molecular formula of C_17_H_17_NO_3._ The structure of **5** was determined to be *N*-*trans*-coumaroyltyramine.

### 2.3. Alpha-Glucosidase Inhibitory Assay

To determined alpha-glucosidase inhibition, the percentage inhibition values at 2 mg/mL were used for the guided isolation, while the half maximal inhibitory concentration (IC_50_) showed the inhibitory ability of compounds (Table 1). The coumarin derivative (**1**) and two tyramine-derived amides (**4** and **5**) exhibited stronger alpha-glucosidase inhibition than the positive control, acarbose.

### 2.4. Enzyme Kinetic Assay

Compounds **1**, **4**, **5**, and acarbose were evaluated their binding mode by the double reciprocal Lineweaver–Burk plot and the inhibition constants (Ki) were determined by secondary plot as shown in Figure 4. Lineweaver–Burk plots of compounds **4** and **5** indicated them to be typical of uncompetitive inhibition with Ki′ values were 51.81 and 1.99 µM, respectively, whereas the plot of **1** displayed mixed-type inhibition. The acarbose standard showed competitive inhibition with Ki values 264.46 µM. Therefore, these could be explained that compounds **4** and **5** were bound with alpha-glucosidase in separated site, compound **1** might both have interacted with the active site and separated site, while acarbose prohibited at the active site of enzyme. For a better understanding of compounds from *N. racemosa*, the molecular docking was performed.

### 2.5. Molecular Docking

Based on the structure-activity relationship, the compounds **1**, **2**, **4**, and **5** that have aromatic part in their structures were evaluated for the protein-ligand complexes. The results indicated that these compounds were bound with the same residues at the entrance area of the active site of alpha-glucosidase, as shown in Table 2 and Figure 5. These findings are in a good agreement with the previous studies, which suggested that the binding site of these compounds is around the active site of this enzyme [9,10,11]. It is well-known that the competitive inhibitor will prohibit the substrate to enter the active site of the enzyme. The docking result showed that compound **1** blocked the alpha-glucosidase entrance area as the line in Figure 5A,B. For an uncompetitive inhibitor, the mechanism of action is different from a competitive inhibitor. An uncompetitive inhibitor does not block the substrate from entering the active site, but rather, prevents the release of the product from the enzyme [11]. As can be seen in Figure 5A,C, the docking results suggest that the two tyramine-derived amides (compounds **4** and **5**) were aligned at the exit part of the enzyme. This alignment would prohibit the release of the product from the alpha-glucosidase. These findings also agreed with previous studies of these two derivatives [12,13]. Moreover, this docking study showed that the binding of compound **2** either prevents the substrate entering the active site or the product leaving the enzyme, as shown in Table 2 and Figure 5A,B. It is consistent with the Costa and team study [9] which suggested that compound **2** might inhibit alpha-glucosidase either in a competitive or an uncompetitive manner.

Two tyramine-derived amides (compounds **4** and **5**) are the promising compounds in the future of the anti-diabetes treatment, since they could strongly inhibit the activity from the alpha-glucosidase and their mode of inhibition (an uncompetitive manner) provided a superior for drug development in the animal model compared with the other modes of inhibition. Even compounds **4** and **5** are highly attractive, but only few studies have examined their structure–activity relationship. Song and team [11] suggested that the hydroxyl group at the ring A (coumaroyl moiety, Figure 5D,E) and α-β unsaturated carbonyl group (Figure 5E) played an important role in the interaction between these two derivatives and alpha-glucosidase. In addition, the substitution of the methoxyl group at the position of C-6 of the ring A caused a significant decrease in the inhibitory activity against the alpha-glucosidase [12]. Therefore, this effect has been used as a reason to explain why compound **4** exhibited less potency than compound **5**. However, it is still unclear how the substitution of this methoxyl group at the position of C-6 impacts the conformation from these compounds at the molecular level and contribute to a better inhibitory activity. Therefore, we evaluated the best poses of compounds **4** and **5** from the molecular docking (as can been seen in Figure 5D,E) and rescoring the obtained binding energy from Autodock Vina using Autodock 4 to obtain more molecular interaction parameters (Table 3). The results indicated that the substitution at C-6 did not only rotate ring A approximately 60 degrees (Figure 5D), but also turned the α-β unsaturated carbonyl group away around 160 degree (Figure 5E). Even this substitution created a great impact in the geometrical structure of these compounds, but these seemed to have less impact in terms of the distance between atoms of these compounds and the residues from the alpha-glucosidase (Table 2). Furthermore, it showed a contradiction in the predicted binding energy showing that compound **4** had a more potent inhibitory activity (lower binding energy) than compound **5** (Table 3). The rescored prediction showed a similar result. However, it pointed out that methoxyl substitution at C-6 of the ring A on compound **4** contributed to higher desolvation and torsion energies (less favorable for enzyme inhibition) than compound **5**, as presented in Table 3. Therefore, these two parameters might be used to explain why compound **5** showed a stronger activity than compound **4**. Although the geometric and energetic changes between compound **4** and **5** from our docking models seem to be inconclusive based on the ligand–enzyme interaction, these alterations have enough impact to cause the difference as high as 30-folds in the anti-glucosidase activity in the in vitro study as mentioned as above.

## 3. Discussion

Alpha-glucosidase is one target enzyme for reduce glucose level in the management of type 2 diabetes mellitus. The *N. racemosa* stem crude extract showed the strong effect towards the enzyme as 96.09% inhibition at 2 mg/mL. For the extraction, step polarity solvents used were increased from *n-*hexane, EtOAc, EtOH, and water, respectively. The bioactivity-guided fractionation indicated that the ethanol extract had the highest activity among these extracts with an IC_50_ of 39.65 μg/mL. The chromatographic techniques conducted the isolation of five compounds (**1**–**5**). Their chemical structures were determined by 1D and 2D NMR analysis. The first time, compound **1** was confusing of its substitution at C-7 and C-8. The NOESY experiment showed the correlation signal between proton of C-6 and methoxyl proton of C-7. Therefore, compound **1** was indicated as scopoletin that is different from iso-scopoletin. Islam and team [14] suggested that the substitution at the 7th and 8th position on the coumarin skeleton results in the alpha-glucosidase inhibitory activity. Compound **1** which has methoxylation and hydroxylation at the 7th and 8th position, respectively, exhibited the alpha-glucosidase more than the compound which does not have these substitutions [14]. This indicated that the substituted group of compound **1** may result to protein interaction. Moreover, another study reported that compound **1** also has the potent inhibitory activity towards another anti-diabetic enzyme, alpha-amylase [15]. Compound **2** was a phenolic compound that was found in various fruits and vegetables and was claimed in therapeutic application for diabetic. The crucial part of **2** was the methoxyl substitution at positions 3 and 5 of the phenol ring [16]. Compound **3** was an α, β-unsaturated ester molecule that has not been reported in regards to diabetic activity. Compounds **4** and **5** were phenylethyl cinnamide derivatives which have a difference of methoxyl substitution. The structure–activity relation study showed that the substitutions of ring A influence the alpha-glucosidase inhibitory activity [11].

The alpha-glucosidase inhibitory activity of isolated compounds, **1**, **4**, and **5** was evaluated as the dose-dependent manner which is shown in Table 1. The coumarin derivative (**1**) and tyramine-derived amides (**4** and **5**) was isolated from many plants [12,13,14,15,17,18,19]. Our study showed the alpha-glucosidase inhibitory activity of **1** as an IC_50_ of 110.97 µg/mL (577.50 µM), while other studies presented various IC_50_ such as 0.057 µM [15], 85.12 µM [18], and 159.16 µM [14]. Compounds **4** and **5** were often isolated from the same plant [12,13], including in our study. In Table 1, the IC_50_ of **4** is 29.87 µg/mL (95.34 µM) and the IC_50_ of **5** is 0.92 µg/mL (3.25 µM). The previous reports showed the IC_50_ of **4** as 3.58 µM [13] and 500.60 µM [12], while the IC_50_ of **5** was presented as 0.40 µM [19], 0.42 µM [17], 0.58 µM [13], and 5.3 µM [12]. The different inhibitory values of the compounds may result from different conditions tested. Besides this, the IC_50_ of compounds **2** and **3** was more than 500 µg/mL. Therefore, our kinetic study focused on in vitro active isolated compounds (**1**, **4** and **5**). The kinetic prediction is that compound **1** was a mixed type inhibitor while tyramine-derived amides (compounds **4** and **5**) inhibited the alpha-glucosidase in an uncompetitive manner. However, the literature reported that compound **1** was a competitive inhibitor towards alpha-glucosidase [20]. In the competitive manner, the inhibitor binds to the free enzyme active site, while the mixed-type manner requires two inhibitors to bind to the enzyme at more than one site [17]. For compounds **4** and **5**, our study agreed with the previous report that showed uncompetitive interaction [11,13]. However, some authors [12] suggested that the mechanisms of compounds **4** and **5** were those of non-competitive inhibitors. The uncompetitive manner is the inhibitor binding to the enzyme–substrate (ES) complex, while the non-competitive manner is the situation that inhibitors bind to both the free enzyme and ES complex with an equivalent affinity [17]. The different study condition may result in the different binding detection, especially in separated sites.

To understand the comprehensive interaction between isolated compounds and alpha-glucosidase, the compounds that have aromatic ring (compounds **1**, **2**, **4** and **5**) were evaluated for the binding site inhibition in molecular level. The results from the molecular docking experiment in this study suggested that these selected bioactive compounds from *N. racemosa* could bind at the same site, the entrance area of the active site, of the alpha-glucosidase. These findings were in a good agreement with the previous independent studies of each compound [9,10,11] and provided comprehensive evidence of the molecular interaction among these compounds and the alpha-glucosidase. Even their modes of inhibition were reported differently, but the results here, in this study, indicated that they could bind exactly at the same binding site. There is a hypothesis that states as a synergistic effect could occur when combining two inhibitors, which had a different mode of inhibition together [21]. Therefore, the question is raised that if it will be the case here with these bioactive compounds from *N. racemosa*. To address this question, further investigations are required. To the best of our knowledge, this was the first study that reported the difference between the molecular conformation from these two-promising alpha-glucosidase inhibitors, namely *N*-*trans*-feruloyltyramine (**4**) and *N*-*trans*-coumaroyltyramine (**5**). Based on the existing knowledge, it was possible to believe that this molecular alteration is caused by the impact of the C-6 substitution of the methoxyl group at the coumaroyl moiety. Therefore, this is the new evidence that showed the effect of this impact at the molecular level. However, it is necessary to perform further experiments for an absolute conclusion.

## 4. Experimental Section

### 4.1. General

Classical column chromatography was used for phytochemical investigation. Silica gel (Vertical, Thailand) and Sephadex^®^ LH-20 (Bio-Sciences, Uppsala, Sweden) were performed as material of normal phase and molecular sieves chromatography, respectively. Thin-layer chromatography (TLC) plate pre-coated with silica gel 60 F_254_ (Merck, Darmstadt, Germany) was used for pretest of purified investigation. TLC spots were detected under daylight, UV light (254 nm and 366 nm) and/or sprayed with 20% H_2_SO_4_. Chromatographic solvents were distilled in our laboratory before used. One-dimensional and two-dimensional NMR spectra were obtained by using a Fourier Transform NMR Spectrometer 500 MHz (Varian, Frankfurt, Germany), in dimethyl sulfoxide (DMSO-*d*_6_) and chloroform (CDCl_3_). UV/Vis spectra were measured on UV-Spectrophotometer Genesys 6 (Thermo, Frankfurt, Germany). IR spectra were determined by Spectrum One FT-IR Spectrometer (PerkinElmer, Buckinghamshire, UK). High resolution electron spray ionization mass spectrometry (HRESIMS) was analyzed by using liquid chromatograph-quadrupole time of fight mass spectrometer (Agilent, CA, USA). The isolated compounds (scopoletin (**1**), syringic acid (**2**), methyl 3-methyl-2-butenoate (**3**), *N-trans*-feruloyltyramine (**4**) and *N-trans*-coumaroyltyramine (**5**)) have purity more than 95%.

### 4.2. Plant Material

The stem of *N. racemosa* was collected from Songkhla province, Thailand, 2014. The herbal material was identified by Assoc. Prof. Dr. Orathai Neamsuvan, Faculty of Traditional Thai Medicine, Prince of Songkla University and the voucher specimen (SKP 054 14 18 01) has been deposited at Department of Pharmacognosy and Pharmaceutical Botany, Faculty of Pharmaceutical Sciences, Prince of Songkla University, Thailand.

### 4.3. Extraction and Isolation

Fresh plant materials (10 kg) were washed and then dried in the oven at temperature 50 °C until dryness. The dried materials (3.1 kg) were macerated with *n-*hexane 3 days in triplicate. The marc and filtrated were separated. The marc was macerated again by the same procedure with ethyl acetate, ethanol and boiled with filtrated water, respectively. Each filtrated solvent was evaporated by reduced pressure at 50 °C. All extracts were kept at 4 °C until used. Due to the bioassay guided isolation, the ethanol crude extract (25.01 g) was subjected to silica flash column using gradient of *n*-hexane to EtOH to achieve 16 fractions (A_1_–A_16_).

Fractions A_7_ and A_8_ were combined and subjected to silica column by using the gradient of mobile phase as CHCl_3_ to CHCl_3_-MeOH (1:1) to give 12 fractions (B_1_–B_12_). Fraction B_4_ and B_5_ were combined and separated on size exclusion method with Sephadex^®^ LH-20 column to give 6 sub-fractions (C_1_–C_6_). C_4_ and C_5_ were combined and further chromatographed on silica column and using the gradient of mobile phase with increasing the polarity by mixing CHCl_3_ to EtOAc and EtOAc-MeOH (9:1) to afford compound **1** (3.9 mg). Fraction B_8_ and B_9_ were combined and subjected to silica column and using the gradient of mobile phase as CH_2_Cl_2_ to EtOAc and EtOAc-MeOH (9:1) to give 15 sub-fractions (D_1_–D_15_). Sub-fractions D_10_ and D_15_ were pooled and washed by EtOAc to yield compound **2** (1.9 mg).

Fraction A_13_ and A_14_ were combined and separated by using silica column and gradient mobile phase as EtOAc to EtOAc-MeOH (1:1) to give 16 sub-fractions (E_1_–E_16_). Sub-fractions E_6_ and E_7_ were combined and subjected to silica column with using gradient mobile phase of CHCl_3_ to CHCl_3_-MeOH (9:1) to achieve compound **3** (1.4 mg).

Fraction A_10_ was chromatographed on silica column and using the gradient mobile phase as *n-*hexane-EtOAc (1:1) to EtOAc-MeOH (9:1) to give 23 fractions (F_1_–F_23_). Fractions F_8_–F_11_ were pooled and partitioned with *n*-hexane, CHCl_3_, and MeOH to yield 3 solvent extracts. The methanolic extract from the partition of F_8_-F_11_ was continuously subjected to silica column and using gradient mobile phase of CHCl_3_ to CHCl_3_-MeOH (9:1) to isolate compound **4** (1.8 mg) and compound **5** (9.9 mg).

### 4.4. Spectroscopic Data

#### 4.4.1. Scopoletin (**1**)

Yellow pale needles; C_10_H_8_O_4_; UV (CHCl_3_) λ_max_ 341 nm; IR ν_max_ cm^−1^: 3460, 1651, 1218, 772, 669; ^1^H NMR (CDCl_3_) δ: 7.58 (1H, d, *J* = 9.5 Hz, H-4), 6.90 (1H, s, H-9), 6.63 (1H, s, H-6), 6.25 (1H, d, *J* = 9.5 Hz, H-3), 6.17 (1H, s, 8-OH), 3.94 (3H, s, 7-OCH_3_); ^13^C NMR (CDCl_3_) δ: 161.43 (C-2), 150.23 (C-10), 149.64 (C-8), 143.96 (C-7), 143.29 (C-4), 113.42 (C-3), 111.48 (C-5), 107.44 (C-6), 103.18 (C-9), 56.39 (CH_3_).

#### 4.4.2. Syringic Acid (**2**)

Orange amorphous solid; C_9_H_10_O_5_; UV (MeOH) λ_max_ 289 nm; IR ν_max_ cm^−1^: 3434, 2066, 1634, 1523, 1468, 1426, 1335, 1281, 1217, 1126, 1015; ^1^H NMR (DMSO-*d*_6_) δ: 7.19 (2H, s, H-2, H-6), 9.07 (1H, brs, 1-COOH), 3.78 (6H, s, 3-OCH_3_, 5-OCH_3_). HR-ESIMS (negative-ion mode) *m/z* 197.0456 [M − H]^−^ (calcd. for [C_9_H_10_NO_5_ − H]^−^).

#### 4.4.3. Methyl 3-Methyl-2-butenoate (**3**)

Brown amorphous solid; C_6_H_10_O_2_; UV (CHCl_3_) λ_max_ 290 nm; IR ν_max_ cm^−1^: 2922, 1698, 1645,1595, 1457, 1322, 1260, 1220, 1110, 874, 800; ^1^H NMR (CDCl_3_) δ: 5.80 (1H, *s*, H-2), 3.8 (3H, *s*, 1-OCH_3_), 1.20 (6H, *s*, 3-CH_3_); ^13^C NMR (CDCl_3_) δ: 186.84 (C-1), 157.30 (C-3), 107.42 (C-2), 56.4 (OCH_3_-1), 29.69 C-4), 29.35 (CH_3_-3).

#### 4.4.4. *N*-*trans*-Feruloyltyramine (**4**)

White amorphous solid; UV (MeOH) λ_max_ 318 nm; IR ν_max_ cm^−1^: 3434, 1652, 1542, 1515, 1457, 1269, 1125, 1032, 978, 819, 670; ^1^H (DMSO-*d*_6_) δ: 7.42 (1H, d, *J* = 15.62 Hz, H-3), 7.11 (1H, d, *J* = 1.95 Hz, H-5), 7.05 (2H, dd, *J* = 9.27, 2.40 Hz, H-4′, H-8′), 7.01 (1H, dd, *J* = 8.29, 1.95 Hz, H-9), 6.78 (1H, d, *J* = 8.06 Hz, H-8), 6.72 (2H, dd, *J* = 9.27, 2.40 Hz, H-5′, H-7′), 6.39 (1H, d, *J* = 15.62 Hz, H-2), 3.87 (3H, s, OCH_3_-6), 3.45 (1H, dd, *J* = 7.08, 7.56 Hz, C-1′), 2.75 (1H, dd, *J* = 7.08, 7.56 Hz, C-2′); ^13^C NMR (DMSO-*d*_6_) δ: 169.19 (C-1), 156.94 (C-6′), 149.88 (C-7), 149.32 (C-6), 142.01 (C-3), 131.33 (C-3′), 130.72 (C-4′, C-8′), 128.31 (C-4), 123.22 (C-9),118.79 (C-2), 116.42 (C-8), 116.28 (C-5′, C-7′), 111.63 (C-5), 56.42 (OCH_3_-6), 42.53 (C-1′), 35.81 (C-2′); HR-ESIMS (positive-ion mode) *m/z* 336.1206 [M + Na]^+^ (calcd. for [C_18_H_19_NO_4_ + Na]^+^).

#### 4.4.5. *N*-*trans*-Coumaroyltyramine (**5**)

White amorphous solid; UV (MeOH) λ_max_ 295 nm; IR ν_max_ cm^−1^: 3434, 1637, 1541, 1456, 1245, 1174, 1105, 980, 830; ^1^H (DMSO-*d*_6_) δ: 7.44 (1H, d, *J* = 15.86 Hz, ah-3), 7.38 (2H, dt, *J* = 9.00, 2.00 Hz, H-4′, H-8′), 7.05 (2H, dt, *J* = 9.00, 2.00 Hz, H-5′, H-7′), 6.78 (2H, m, H-5, H-9), 6.70 (2H, m, H-6, H-8), 6.38 (1H, d, *J* = 15.86 Hz, H-2), 3.45 (2H, dd, *J* = 7.07, 7.57 Hz, H-1′), 2.75 (2H, dd, *J* = 7.57, 7.08 Hz, H2′); ^13^C NMR (DMSO-*d*_6_) δ: 169.25 (C-1), 160.50 (C-7), 156.92 (C-6′), 141.76 (C-3), 131.34 (C-3′), 130.72 (C-4′, C-8′), 130.53 (C-5, C-9), 127.77 (C-4), 118.49 (C-2), 116.72 (C-5′, C-7′), 116.28 (C-6, C-8), 42.53 (C-1′), 35.82 (C-2′); HR-ESIMS (positive-ion mode) *m/z* 306.1101 [M + Na]^+^ (calcd. for [C_17_H_17_NO_3_ + Na]^+^).

### 4.5. Enzymatic Assay

The alpha-glucosidase inhibitory assay was slightly modified from Dej-adisai and Pitakbut, 2015 [22]. This assay was determined by a colorimetric method which observed the yellow product, *p*-nitrophenol (*p*NP), and detected by visible light at 405 nm with a SPECTROstar Nano spectrophotrometer (BMG Labtech, Ortenberg, Germany). The UV absorbance of the final product, *p*NP, was converted to velocity (V) by the following equation: Velocity = ∆ Absorbance at 405 nm/∆ Time(1)
while the velocity was calculated percentages of inhibition as in the following equation: % Inhibition = [(V_control_ − V_sample_)/V_control_] × 100(2)

The IC_50_ value that performed the inhibition ability of samples was received from the calibration curve plotted between percentages of inhibition and five concentrations of samples.

### 4.6. Enzyme Kinetic Study

The double reciprocal Lineweaver–Burk plot was used to determine the mode of inhibition, while the secondary plots conducted the inhibition constants. The inhibition constant was showed as Ki values when the inhibitor binds with free enzyme for competitive inhibition, but it was showed as Ki′ values when the inhibitor binds with the enzyme–substrate complex for uncompetitive inhibition. Briefly, the enzyme inhibition procedure was performed as mention above. The 6 concentrations of *p*NPG (5–0.1 mM), the fixed enzyme concentration (1 unit/mL), and the 3 concentrations of each effective sample were evaluated for the manner of inhibition. 

### 4.7. Molecular Docking

Autodock Vina version 1.1.2 from the Scripps Research Institute, San Diego, California, USA was used to investigate the molecular interaction in this study [9]. The target protein, the alpha-glucosidase (PDB ID: 3a4a), was downloaded from RCSB Protein Data Bank (https://www.rcsb.org/, accessed on 1 May 2021), while the compounds of interest were downloaded from Pubchem (https://pubchem.ncbi.nlm.nih.gov/, accessed on 1 May 2021). Before performing the experiment, both the alpha-glucosidase and the compounds of interest needed to be prepared properly. Firstly, Autodock Tool version 1.5.6 from the same institute as Autodock Vina was used to prepare the alpha-glucosidase and to identify the active site of this enzyme by using the native ligand, the glucose molecule, as a guideline [23]. The center of this active site was determined and presented as the three-dimensional grid, which X = 21.1, Y = −7.4 and Z = 24.2, respectively. This grid had a size of 17 Å × 17 Å × 17 Å. Secondly, Avogadro Version 1.2.0 was used to perform the two steps energy minimization for all the compounds in this study. The first step performed a geometric optimization and the following by a general amber force field, GAFF [24]. To perform this experiment, all docking parameters must be defined and almost all these parameters were set as a default. Only the exhaustiveness value was adjusted to 24. Importantly, this protocol was validated by re-docking the native ligand of the glucose molecule into the identified active site of this α-glucosidase under the setup condition, as described earlier, before applying this protocol in this study. As expected, the re-docking result passed the acceptance criterion, since the RMSD was less than 1 Å [25]. The trial result would not be presented here, but rather, in the supplementary file. Therefore, this protocol is reliable and ready for the experiment. For post-docking analysis, Chimera version 1.11.2 together with viewdock package [26] was used to visualize and evaluate the outcomes from this experiment. Finally, we rescored the binding energy of the best docking pose using Autodock 4.2.6, following previous reports [25,27].

## 5. Conclusions

This study would be the first report on the phytochemical investigation of *Neuropeltis racemosa* which is used as the component of many traditional Thai recipes. The bioassay-guided fractionation on alpha-glucosidase inhibition revealed that the best active extract is the ethanolic extract. The extracts were separated by using classical column chromatographic techniques and five compounds were obtained. The spectroscopic techniques including UV-Vis, FTIR, HRMS, and NMR (^1^H-NMR, ^13^C-NMR, HMQC, HMBC, NOESY) analyses revealed the chemical structure of the isolated compounds. Then, the compounds were estimated for their bioactive potential of alpha-glucosidase inhibition. Moreover, three compounds (compound **1**, **4**, and **5**) which have the better activity than the standard acarbose were chosen for the mechanism of action analysis. Some of them exhibited the significant alpha-glucosidase inhibition with different mechanism of action. Besides this, an in silico study exposed the alpha-glucosidase inhibition of selected compounds (compound **1**, **2**, **4**, and **5**) at the molecular level. The parameters of molecular docking such as binding energy, desolvation, and torsion energy were used for the explanation of the molecular interaction. Based on the results, these findings justify the value of *N. racemosa* as a component of traditional medicine and resource of lead antidiabetic compounds for further study.

## Figures and Tables

**Figure 1 molecules-27-00639-f001:**
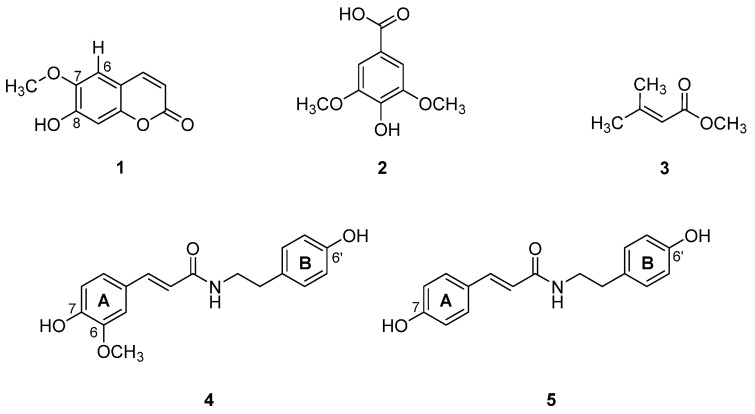
Isolated compounds (**1**–**5**) from *N. racemosa* stem.

**Figure 2 molecules-27-00639-f002:**
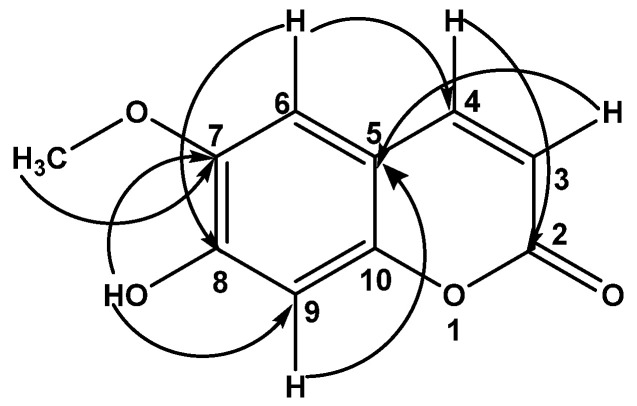
HMBC correlations (from H to C) of compound **1**.

**Figure 3 molecules-27-00639-f003:**
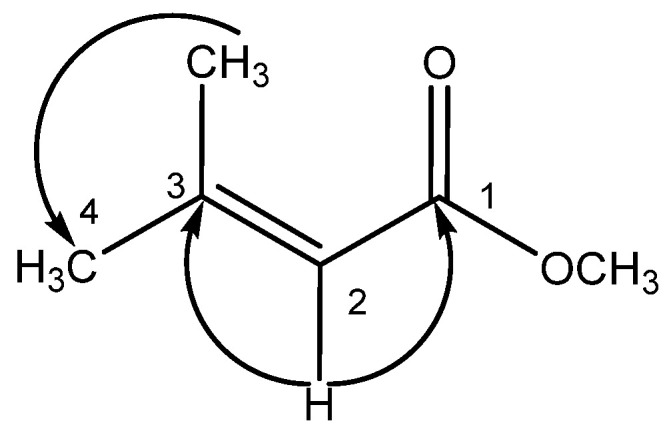
HMBC correlations (from H to C) of compound **3**.

**Figure 4 molecules-27-00639-f004:**
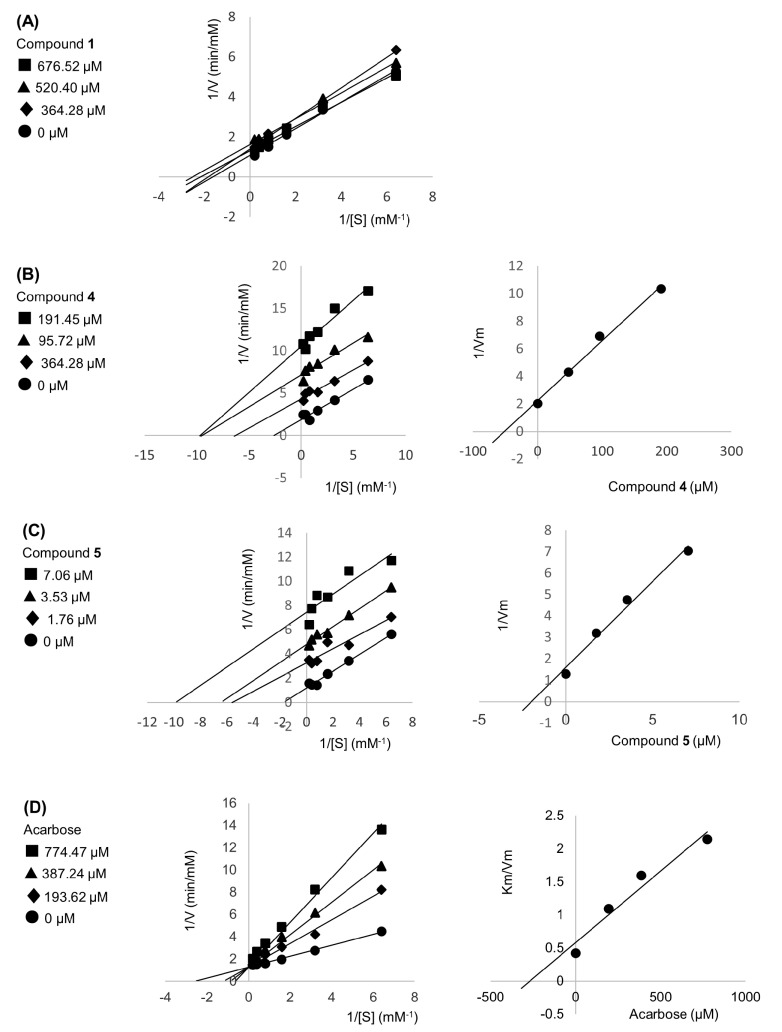
Lineweaver−Burk plots of compound **1**, **4**, **5**, and acarbose, respectively (**A**–**D**). The secondary plots of each compound are on the right.

**Figure 5 molecules-27-00639-f005:**
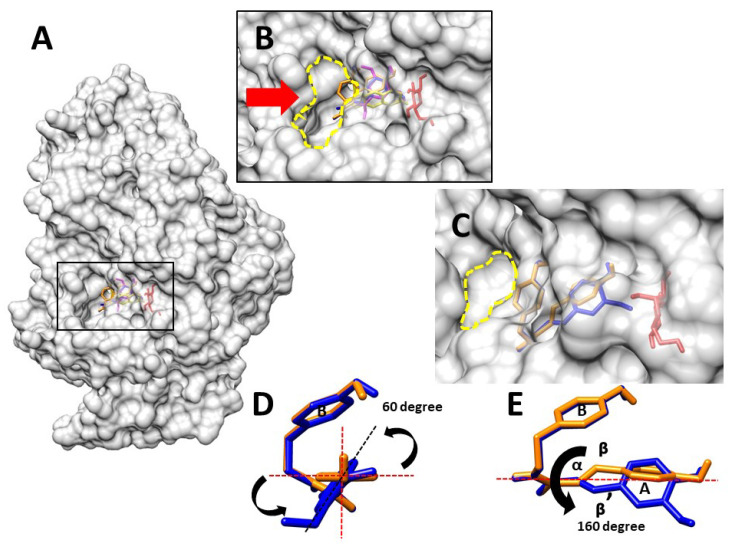
The molecular interaction between the alpha-glucosidase, presented in grey color and glucose as in red color, **1** as in yellow color, **2** as in pink color, **4** in blue color and **5** as in orange color, respectively. (**A**) The molecular interaction between the alpha-glucosidase and the mentioned. (**B**) The expanded picture at the active site of the alpha-glucosidase. The dashed circle in yellow color indicated the entrance gate to the active site, while the red arrow was used to emphasis this gate. (**C**) The comparison between the docked conformations of **4** in blue color and **5** as in orange color at the entrance gate of the active site. (**D**,**E**) The alteration of the docked conformation between **4** as in blue color and **5** as in orange color. The red dashed lines indicated the X and Y-axis, while the black arrows indicated the changing of the position in the chemical structure.

**Table 1 molecules-27-00639-t001:** Inhibitory activity of isolated compounds from *N. racemosa* on alpha-glucosidase.

Compounds	IC_50_ (µg/mL)	Inhibition Type	Ki ^b^ or Ki′ ^c^ (µM)
Scopoletin (**1**)	110.97	Mixed	-
Syringic acid (**2**)	>500	N.T. ^a^	-
Methyl 3-methyl-2-butenoate (**3**)	>500	N.T. ^a^	-
*N-trans*-feruloyltyramine (**4**)	29.87	Uncompetitive	51.81 ^c^
*N-trans*-coumaroyltyramine (**5**)	0.92	Uncompetitive	1.99 ^c^
Acarbose (Positive control)	272.72	Competitive	264.46 ^b^

^a^ Not tested, ^b^ Inhibition constant when inhibitor bound with free enzyme, ^c^ Inhibition constant when inhibitor bound with enzyme–substrate complex.

**Table 2 molecules-27-00639-t002:** Molecular docking interaction of the alpha-glucosidase (AR) and selected compounds.

Complex	Number(s) of Interaction	Interaction Sites	Distances (Å)
AR—**1**	1	Glucose 601	**1**—Glucose 601 (3.41 Å)
AR—**2**	4	Tyr 158	**2**—Tyr 158 (2.05 Å)
		Glucose 601	**2**—Glucose 601 (1.36 Å)
			**2**—Glucose 601 (3.14 Å)
			**2**—Glucose 601 (4.21 Å)
AR—**4**	4	Arg 315	**4**—Arg 315 (4.60 Å)
		Asn 415	**4**—Asn 415 (4.18 Å)
		Glucose 601	**4**—Glucose 601 (4.33 Å)
			**4**—Glucose 601 (3.21 Å)
AR—**5**	4	Arg 315	**5**—Arg 315 (4.64 Å)
		Asn 415	**5**—Asn 415 (4.28 Å)
		Glucose 601	**5**—Glucose 601 (4.14 Å)
			**5**—Glucose 601 (2.88 Å)

**Table 3 molecules-27-00639-t003:** Rescoring binding energy of the interested compounds (**1** to **5**) from Autodock 4.2.6 compared to Autodock Vina.

Autodock 4.2.6									Autodock Vina
Compound	vdW+Hbond (I)	Elec. Energy (II)	Desol. energy (III)	Total Intermol. Interact. Energy (IV; I+II+III)	Total Internal Energy (V)	Tors. Free Energy (VI)	Unbound’s Energy (VII)	Binding Energy (Kcal/mol) (VIII; IV+V+VI+VII)	Affinity (Kcal/mol)
**1**	−6.34	0.01	1.80	−4.53	−0.69	0.6	0	−4.62	−6.3
**2**	−6.67	1.70	3.00	−1.98	−0.58	1.49	0	−1.07	−5.7
**4**	−9.27	0.30	2.60	−6.36	−1.45	2.39	0	−5.42	−7.5
**5**	−8.22	0.29	2.05	−5.88	−1.36	2.09	0	−5.15	−7.0

vdW+Hbonding = Van der Waals + Hydrogen bonding, Elec. energy = Electrostatic energy, Desol. energy = Desolvation energy, Total Intermol. Interact. energy = Total Intermolecule Interaction energy, Tors. free energy = Torsion free energy. Highlighted color = higher energy of compound **5** than compound **4**.

## Data Availability

Data sharing is not applicable.

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
