# Peer review of "Alpha-Glucosidase Inhibition and Molecular Docking of Isolated Compounds from Traditional Thai Medicinal Plant, Neuropeltis racemosa Wall."

_molecules, 2022, doi:10.3390/molecules27030639_

Round 1

Reviewer 1 Report

This manuscript is quite interesting and has comprehensive scientific value starting from extraction, determining the chemical structure of isolate compounds, determining activity and tracing the interaction model of isolate compounds with target receptors. However, there are two things that need to be added to strengthen the scientific rationale:

  1. The binding score should be added in Table 2. It will describe the strength of the binding between the isolate compound and the receptor
  2. In lines 221-222 there is a confusion. Based on the data in table 1, IC50 of compound 4 is 29.87 microgram/mL, while compound 5 is 0.92 microgram/mL. So, compound 5 is more potent again enzyme than compound 4. Please check again in the manuscript.

Please check the grammar error using available program.

Author Response

Comments and Suggestions for Authors

This manuscript is quite interesting and has comprehensive scientific value starting from extraction, determining the chemical structure of isolate compounds, determining activity and tracing the interaction model of isolate compounds with target receptors. However, there are two things that need to be added to strengthen the scientific rationale:

  1. The binding score should be added in Table 2. It will describe the strength of the binding between the isolate compound and the receptor

### Thank you for your kind suggestion. We add the binding score in the new Table 3 and also add the explanation of binging energy result in this manuscript.

  1. In lines 221-222 there is a confusion. Based on the data in table 1, IC50 of compound 4 is 29.87 microgram/mL, while compound 5 is 0.92 microgram/mL. So, compound 5 is more potent again enzyme than compound 4. Please check again in the manuscript.

### We apologize for the mistake. However, we have already corrected the new message as “Therefore, this effect has been used as a reason to explain why compound 4 exhibited the less potency than compound 5.” in this manuscript.

Please check the grammar error using available program.

We tried to edit it already.

Finally, we would like to thank for your kind suggestion and hope that our revised manuscript will be satisfy for you to accept in order to publish in this journal.

Best regards,

Sukanya and team

14 Jan 2022

Reviewer 2 Report

The manuscript described the bioactive-guided isolation of Neuropeltis racemosa, leading to five compounds. The NMR data and elucidation of compounds are correct. The alpha-glucosidase inhibition of compounds 1,4, and 5 was evaluated, indicating that they are strong inhibitors. However, there are some concerns for the manuscript:

Compounds 1, 4, and 5 were common in higher plants, and their alpha-glucosidase inhibition has been studied in many papers. The manuscript lacked the discussion to compare the current results with those in the literature regarding alpha-glucosidase inhibition of 1, 4, and 5.

For 4 and 5, there are two references having the same content: "Liu et al, Nat. Prod. Comm. 6, 851-853, 2011" and "Panidthananon et al., Molecules, 23, 1600". However, the results from the reference "Liu et al. 2011" and the current data are opposite. Please discuss and explain. Please revise the sentence in lines 264-265 about the consistency of two data.

Compound 1 was deeply studied for in vitro alpha-glucosidase inhibition and anti-diabetic. Please discuss more in the manuscript.

Please provide IC50 values of extracts. The inhibition percent is not enough to determine the most active extract because the tested concentrations were very high (2 mg/ml).

Please provide the purity of compounds.

Please rewrite the conclusion part. It is too short and did not summarize all data.

Minor points:

Please delete all diameter and length of columns in the experimental part.

"hexane" change to "n-hexane"

"N-trans-...." to "N-trans-....". Please check and change thoroughly.

L319: Fractions A7 and A8... Similar for L329/L334

Author Response

Comments and Suggestions for Authors

The manuscript described the bioactive-guided isolation of Neuropeltis racemosa, leading to five compounds. The NMR data and elucidation of compounds are correct. The alpha-glucosidase inhibition of compounds 1,4, and 5 was evaluated, indicating that they are strong inhibitors. However, there are some concerns for the manuscript:

Compounds 1, 4, and 5 were common in higher plants, and their alpha-glucosidase inhibition has been studied in many papers. The manuscript lacked the discussion to compare the current results with those in the literature regarding alpha-glucosidase inhibition of 1, 4, and 5.

### Yes, we add more discussion in the manuscript.

The coumarin derivative (1) and tyramine-derived amides (4 and 5) was isolated from many plants [12-15, 17-19]. Our study showed the alpha-glucosidase inhibitory activity of 1 as IC50 110.97 µg/ml (577.50 µM), while other studies presented with various IC50 such as 0.057 µM [15], 85.12 µM [18], and 159.16 µM [14]. Oftentimes compounds 4 and 5 were isolated from the same plant [12, 13] including our study. In table 1, the IC50 of 4 is 29.87 µg/ml (95.34 µM) and the IC50 of 5 is 0.92 µg/ml (3.25 µM). The previous reports showed IC50 of 4 as 3.58 µM [13] and 500.60 µM [12], while IC50 of 5 was presented as 0.40 µM [19], 0.42 µM [17], 0.58 µM [13] and 5.3 µM [12]. The different inhibitory values of compounds may result from different condition tested.

For 4 and 5, there are two references having the same content: "Liu et al, Nat. Prod. Comm. 6, 851-853, 2011" and "Panidthananon et al., Molecules, 23, 1600". However, the results from the reference "Liu et al. 2011" and the current data are opposite. Please discuss and explain. Please revise the sentence in lines 264-265 about the consistency of two data.

### We apologize for the mistake. However, we add another reference that show the consistency result with our study and we also discuss to experimental situation as your suggestion in the manuscript.

For compounds 4 and 5, our study agreed with the previous report that showed uncompetitive interaction [11, 13]. However, some literature [12] suggested the mechanism of compounds 4 and 5 were non-competitive inhibitors. The uncompetitive manner is the inhibitor binding to the enzyme-substrate (ES) complex, while non-competitive manner is the situation that inhibitors bind to both free enzyme and ES complex with equivalent affinity [17]. The different study condition may result to the different binding detection, especially separated sites.

Compound 1 was deeply studied for in vitro alpha-glucosidase inhibition and anti-diabetic. Please discuss more in the manuscript.

### Yes, we add more discussion in the manuscript.

Islam and team [14] suggested that the substitution at the 7th and 8th position on coumarin skeleton result to the alpha-glucosidase inhibitory activity. Compound 1 which has methoxylation and hydroxylation at 7th and 8th position, respectively exhibited the alpha-glucosidase more than the compound which does not have these substitutions [14]. This indicated that the substituted group of compound 1 may result to protein interaction. Moreover, other study has been reported that compound 1 also exhibited the potent inhibitory activity to another anti-diabetic enzyme, alpha-amylase [15].

Please provide IC50 values of extracts. The inhibition percent is not enough to determine the most active extract because the tested concentrations were very high (2 mg/ml).

### Yes, we have already provided IC50 values of extract as your suggestion already in result part.

n-Hexane, EtOAc, EtOH and water extracts exhibited alpha-glucosidase inhibition with IC50 56.81 g/ml, 191.44 μg/ml, 39.65 μg/ml, 4.02 mg/ml, respectively while the standard, acarbose exhibited IC50 245.95 μg/ml.

Please provide the purity of compounds.

### Yes, we have already provided the purity of compounds in topic general of experimental section.

Please rewrite the conclusion part. It is too short and did not summarize all data.

### Yes, we have already rewritten the conclusion part.

This study would be the first report on phytochemical investigation of Neuropeltis racemosa which is used as the component of many traditional Thai recipes.  The bioassay-guided fractionation on alpha-glucosidase inhibition revealed the best active extract is the ethanolic extract. The extracts were separated by using classical column chromatographic techniques were obtained 5 compounds. The spectroscopic techniques including UV-Vis, FTIR, HRMS and NMR (1H-NMR, 13C-NMR, HMQC, HMBC, NOESY) analyses revealed the chemical structure of the isolated compounds. Then, the compounds were estimated their bioactive potential on alpha-glucosidase inhibition. Moreover, 3 compounds (compound 1, 4 and 5) which have the better activity than the standard acarbose were chosen for the mechanism of action analysis. Some of them exhibited the significant alpha-glucosidase inhibition with different mechanism of action. Besides of this, in silico study exposed alpha-glucosidase inhibition of selected compounds (compound 1, 2, 4 and 5) in the molecular level. The parameters of molecular docking such as binding energy, desolvation and torsion energy were used for the explanation of molecular interaction. Base on the results, these finding justify the value of N. racemosa as component of traditional medicine and resource of lead antidiabetic compounds for further study.

Minor points:

Please delete all diameter and length of columns in the experimental part.

### Yes, we already deleted all diameter and length of columns in the experimental part.

"hexane" change to "n-hexane"

### Yes, we have already changed "hexane" to "n-hexane".

"N-trans-...." to "N-trans-....". Please check and change thoroughly.

### Yes, we have already changed "N-trans-...." to "N-trans-....".

L319: Fractions A7 and A8... Similar for L329/L334

### Yes, we have already added the words “Fractions” on line as your suggestion.

Finally, we would like to thank for your kind suggestion and hope that our revised manuscript will be satisfy for you to accept in order to publish in this journal.

Best regards,

Sukanya and team

14 Jan 2022

Round 2

Reviewer 2 Report

The remarks were revised accordingly.